# Cyberaggression in Adolescents of Bolivia: Connection with Psychopathological Symptoms, Adaptive and Predictor Variables

**DOI:** 10.3390/ijerph17031022

**Published:** 2020-02-06

**Authors:** Maite Garaigordobil, Juan Pablo Mollo-Torrico, Juan Manuel Machimbarrena, Darío Páez

**Affiliations:** Faculty of Psychology, University of the Basque Country UPV/EHU, Avenida de Tolosa, 70, 20018 Donostia, Spain; juanpablomollot@icloud.com (J.P.M.-T.); juanmanuel.machimbarrena@ehu.eus (J.M.M.); dario.paez@ehu.eus (D.P.)

**Keywords:** cyberbullying, cyberaggression, prevalence, psychopathology, self-image, empathy, happiness, adolescence, Bolivia, predictors

## Abstract

Concern about the increase of cyberbullying underlies this study, which had four objectives: (1) to calculate the prevalence of cyberaggressors; (2) to compare non-cyberaggressors with cyberaggressors in other bullying/cyberbullying roles, in psychopathological symptoms, and in self-image of masculinity/femininity, happiness, and empathy; (3) to analyze whether cyberaggressors consulted with a psychologist more than non-cyberaggressors; and (4) to identify predictor variables of cyberaggression. Participants were 1558 Bolivian students aged 13 to 17 years. Seven evaluation instruments were administered, using a descriptive, comparative, cross-sectional methodology. Results: (1) 32.7% of cyberaggressors (27.4% occasional, 5.3% severe) were found, with a higher percentage of males; (2) compared to non-cyberaggressors, cyberaggressors engaged in more face-to-face bullying behaviors, suffered more face-to-face victimization and cybervictimization, had more psychopathological symptoms (depression, somatization, obsession-compulsion, interpersonal sensitivity, anxiety, hostility, paranoid ideation, psychoticism), higher overall level of psychopathology, had requested psychological assistance in a greater proportion, self-defined with many attributes associated with masculinity, and felt less happiness and less empathy; and (3) being or having been a cybervictim, being or having been an aggressor of face-to-face bullying, low empathetic joy, and a self-image based on attributes associated with masculinity were predictors of cyberaggression. The need for therapeutic intervention with all those involved and the importance of prevention in the school context are discussed.

## 1. Introduction

Cyberbullying poses a threat to mental health and psychological well-being during childhood, adolescence, and youth. Cybervictims suffer feelings of insecurity, loneliness, sadness, unhappiness, helplessness, anxiety, irritability, depression, suicidal ideation (some commit suicide), post-traumatic stress, fear, low self-esteem, anger and frustration, somatizations, sleep disorders, eating disorders, phobias, academic performance issues etc. In addition, these consequences are often long-lasting in the medium and long terms. Cyberaggressors are more likely to show moral disengagement, lack of empathy, low emotional stability, difficulty to follow rules, delinquent behavior, problems with aggressive behavior, dependence on technologies, school absenteeism, alcohol and drug intake, feelings of emotional loneliness, less optimism and happiness etc. (for a review, see Garaigordobil [1]). The prevalence of bullying in any of its modalities and its serious consequences reveal the scope of the public health problem involved. 

Whereas the study of victimization has received considerable attention, there is less research on cyberaggression, its connections with other variables (psychopathological and adaptive), and/or the variables that can predict this type of digital aggressive behavior. Therefore, the main objective of this study revolves around cyberaggression. Its prevalence is explored in a sample of adolescents of Bolivia, the connections of cyberaggression with psychopathological symptoms, with adaptive variables, the demand for psychological assistance, and the variables that predict these behaviors.

### 1.1. Cyberaggression: Prevalence and Sex Differences

The systematic review of studies carried out on the prevalence of bullying and cyberbullying in Latin America published between 2005 and 2018 [2] showed a high prevalence of cyberaggressors, whose percentages in the different studies range from 2.5% to 32%. The findings confirm a significant prevalence of cyberaggressors in all geographical, cultural, and educational contexts of Latin America. Bolivia is one of the countries in Latin America with high prevalence of bullying and cyberbullying, although only the study of Egüez and Schulmeyer [3] provided information on cyberaggressors, finding 12%. On the other hand, most studies that have explored sex differences find a higher percentage of male cyberaggressors, although the sex differences in online bullying are decreasing [1,4,5,6,7].

### 1.2. Cyberaggression and Psychopathology 

Concerning the connections between cyberaggression and psychopathology, the studies show that cyberaggressors suffer from depression and anxiety [8,9], stress [10], feelings of emotional loneliness [11,12], psychosomatic problems, and psychopathological symptoms [13]. The study of Garaigordobil and Machimbarrena [14] found that those who had high scores in cyberaggression also had high levels of stress, emotional and behavioral problems, and had consulted a psychologist for various symptoms (depression, anxiety etc.) more than those who were not involved in situations of cyberbullying. The study of Cañas, Estévez, Marzo, and Piqueras [15] also found that, compared to non-cyberaggressors, severe cyberaggressors showed increased perceived stress, loneliness, and depression.

### 1.3. Cyberaggression: Connection with Self-Image Associated with Gender Stereotypes, Happiness, and Empathy

No studies exploring the self-image of cyberaggressors associated with gender stereotypes (masculinity–femininity) have been found in the review. 

The relationship between cyberaggression and happiness has received little attention. In the study of Navarro and collaborators [16], being a cyberaggressor was associated with less optimism and global happiness and, in the work of Cañas et al. [15], severe cyberaggressors were less satisfied with life, compared to non-cyberaggressors.

In general, the studies find that, compared to non-cyberaggressors, cyberaggressors have lower empathy [17,18], but the studies that analyze cognitive empathy and affective empathy separately show mixed results. Renati and collaborators only found less emotional empathy [19], but in the study of Rodríguez-Hidalgo and collaborators, low affective and cognitive empathy were predictors of cyberaggression [20]. 

### 1.4. Predictor Variables of Cyberaggression

Some research has shown the connections between cyberaggression and other bullying roles (victimization and perpetration) and cybervictimization, as well as the predictor value of these roles. Specifically, cyberaggression in the 9th grade was predicted by relational aggression (spreading rumors about someone, excluding others etc.) performed in the 7th grade [21]. Another study showed that antisocial behavior (traditional bullying and rule breaking) predicted cyberaggression [22]. Participating in traditional bullying is an additional risk factor, as bullying aggressors also tend to be cyberaggressors [23], and even having been a perpetrator of face-to-face bullying was a predictor of cyberaggression [24]. Several studies found that having been a victim and an aggressor of bullying were predictors of cyberaggression [25,26], and the meta-analysis of Guo [27] confirmed this, emphasizing that having been a bullying aggressor was a robust predictor of being a cyberaggressor. In addition, an association was identified between being a cyberaggressor and being a cybervictim [28], and even having been cybervictimized was a predictor of cyberaggression [20,24].

### 1.5. Objectives and Hypotheses

From a cognitive behavioral theoretical framework, and due to our concern about the increase of cyberbullying, this study had four objectives: (1) to calculate the prevalence of cyberaggressors, the percentage of students who have performed behaviors of cyberbullying (occasional and severe cyberaggressors), exploring possible sex differences and the most prevalent behaviors; (2) to compare non-cyberaggressors (adolescents who have never engaged in cyberbullying behavior), with cyberaggressors (who have performed occasional or frequent cyberbullying behaviors) in other bullying/cyberbullying roles (victim, aggressor, cybervictim), in psychopathological symptoms (depression, social anxiety, somatization, obsession-compulsion, interpersonal sensitivity, phobic anxiety, paranoid ideation, psychoticism), in self-image of masculinity/femininity, and happiness and empathy; (3) to analyze whether cyberaggressors have consulted a psychologist significantly more frequently than non-cyberaggressors, due to psychological distress and diverse symptoms (anxiety, depression etc.); and (4) to identify variables that predict cyberaggression. With these objectives, and after the literature review, this study proposes four hypotheses:

**H1.** *Taking into account the prevalence found in the review of Latin American studies [2], it is expected to find approximately 30% of cyberaggressors (who have performed one or more cyberbullying behaviors) of whom 5% will be severe (have very frequently performed cyberbullying behaviors)*.

**H2.** *Cyberaggressors (occasional and severe) are more involved in bullying/cyberbullying situations (as victims, aggressors, and cybervictims), and show more psychopathological symptoms (depression, social anxiety, somatization, obsession-compulsion, interpersonal sensitivity, anxiety, hostility, phobic anxiety, paranoid ideation and psychoticism), a higher level of general psychopathology, a self-image based on many attributes associated with masculinity and few associated with femininity, low happiness, and low empathy*.

**H3.** *Cyberaggressors seek psychological attention significantly more frequently than individuals who do not engage in cyberbullying behaviors towards other classmates*.

**H4.** *Having been an aggressor in face-to-face bullying, having suffered cybervictimization, and low empathy are predictors of cyberaggression*.

## 2. Materials and Methods 

### 2.1. Participants

The study sample is made up 1558 adolescents (50.2% girls, 49.8% boys) of Cochabamba (Bolivia) aged 13 to 17 years (mean age = 14.64, standard deviation = 0.96) from 18 schools. Cochambaba is a city of Bolivia (population *n* = 1,916,000) with 195 centers of secondary education. Concerning educational level, 53.7% are in 3rd grade of Secondary Education and 46.3% are studying 4th grade (54.9% public schools and 45.1% in private schools). The sample was selected randomly and is representative in of the students of the last cycle of Secondary Education of the Cochabamba (*n* = 31.895). Using a confidence level of 0.99, with a sample error of 0.03%, the representative sample is 1500. A stratified sampling technique was used to select the sample, taking into account the following parameters: type of school (public-private), and educational level (3rd and 4th grades). 

### 2.2. Measuring Instruments

To measure the target variables, we used 7 standardized instruments with psychometric guarantees of reliability and validity.

*Cyberbullying: Screening of Peer Harassment* [29,30]. This is a standardized instrument to assess bullying and cyberbullying behavior. The Bullying Scale assesses four types of face-to-face bullying: physical, verbal, social, and psychological. The Cyberbullying Scale explores 15 cyberbullying behaviors such as: sending offensive and insulting messages, making offensive calls, recording a beating and uploading it to YouTube, disseminating compromising photos or videos, stealing and disseminating photos, making anonymous frightening calls, blackmailing, or threatening someone, sexual harassment, spreading rumors, secrets, and lies, stealing email passwords, faking photos or videos and uploading them to YouTube, isolating others from social networks, blackmailing with disclosing intimate details about someone, death threats, slandering. Adolescents report the frequency with which they have suffered and performed these behaviors in the course of their lives. Each behavior is scored as 0 (*never*), 1 (*sometimes*), 2 (*fairly often*), or 3 (*always*). The test provides a global level of victimization and aggression on both scales. The four indicators report the amount of behavior suffered and performed from both face-to-face bullying and cyberbullying. The psychometric studies confirm adequate internal consistency both in the bullying scale (α = 0.81) and the cyberbullying scale (α = 0.91), in the same direction as those obtained with the sample of this study (bullying α = 0.78; cyberbullying α = 0.87). The alpha indices of the subscales were as follows: 0.73 for bullying victimization; 0.79 for bullying perpetration; 0.83 for cyberbullying victimization and; 0.88 for the cyberbullying perpetration subscale. 

*Beck Depression Inventory-II* (BDI-II) [31,32]. This inventory is composed of 21 items that measure the severity of depression. The items measure symptoms of depression: sadness, pessimism, feelings of failure, loss of pleasure, feeling guilty, feelings of punishment, self-dissatisfaction, self-criticism, thoughts of suicide, crying, agitation, loss of interest, indecision, futility, loss of energy, changes in sleep pattern, irritability, changes in appetite, difficulty concentrating, tiredness or fatigue, and loss of interest in sex. The adolescent reports the degree to which he or she has had these symptoms over the past two weeks. The alpha coefficients obtained with the original sample for students were high (α = 0.93), as in the Spanish adaptation (α = 0.87), and in the sample of this study (α = 0.92).

*Social Anxiety Scale for Adolescents* (SAS-A) [33,34]. This is made up of 22 items that evaluate global social anxiety (social phobia) and 3 sub-dimensions: fear of negative evaluation, social avoidance and distress in the face of unknown situations and strangers, and stress in the company of acquaintances. Adolescents report how often (*never*-*always*) they have such thoughts, feelings, behaviors etc. The internal consistency obtained in the Spanish adaptation with adolescent sample was very high (α = 0.91), as in the sample in this study (α = 0.91).

*Symptoms Checklist-90-Revised (SCL-90-R)* [35,36]. In this study, 83 items distributed on 9 scales were administered, which report the psychopathological disorders: somatization (experiences of body dysfunction, neurovegetative alterations of the cardiovascular, respiratory, gastrointestinal and muscular systems), obsession-compulsion (absurd and unwanted behaviors, thoughts etc. that generate intense distress and are difficult to resist, avoid, or eliminate), interpersonal sensitivity (timidity and embarrassment, discomfort and inhibition in interpersonal relationships), depression (anhedonia, hopelessness, helplessness, lack of energy, self-destructive ideas etc.), anxiety (generalized and acute anxiety/panic), hostility (aggressive thoughts, feelings and behaviors, anger, irritability, rage, and resentment), phobic anxiety (agoraphobia and social phobia), paranoid ideation (paranoid behavior, suspicion, delirious ideation, hostility, grandiosity, need for control etc.), and psychoticism (feelings of social alienation). Furthermore, the test makes it possible to calculate the General Symptomatic Index (GSI), which is a standard and indiscriminate measure of the intensity of global psychosomatic and psychic suffering. Adolescents report the frequency with which they have experienced these symptoms during the last month. Studies with Spanish samples suggest good reliability (α = 0.81 to 0.90), as in this study (α = 0.92). Specifically, the reliability of each subscale was the following: somatization (α = 0.92); obsession-compulsion (α = 0.92); interpersonal sensitivity (α = 0.91); depression (α = 0.92); anxiety (α = 0.94) hostility (α = 0.90); phobic anxiety (α = 0.90); paranoid ideation (α = 0.89) and; psychoticism (α = 0.92).

*Bem Sex Role Inventory (BSRI)* [37,38]. The purpose of the test is to evaluate people’s identification with gender role characteristics, called masculinity and femininity. These characteristics emphasize achievement, materialism, and competition or affective sharing, quality of life, and interpersonal harmony. Bem states that these two constructs are two different dimensions that can be found in the same person. The test is administered to identify the self-image or self-concept of femininity and masculinity of each adolescent. The inventory features 18 adjectives that define self-image. The adolescents report the degree to which these adjectives define them, on a Likert scale ranging from 1 (*not at all*) to 7 (*a lot*). Items of the masculinity scale are: athletic, strong personality, wants to take risks/loves danger, dominant, aggressive, acts as leader, individualist, hard, and selfish. Items of the femininity scale are: affectionate, sensitive to the needs of others, understanding, compassionate, warm, tender, child-loving, weeps easily, submissive. The inventory has good internal consistency (α = 0.80) in the same direction as that obtained in the Spanish adaptation and with the sample of this study (α = 0.85). Specifically, the alpha coefficient for the masculinity scale was 0.77 and 0.82 for the femininity scale.

*The Oxford Happiness Questionnaire* (OHQ) [39,40]. The OHQ was derived from the Oxford Happiness Inventory, which reduced 29 items, attempts to measure the happiness of a general nature of each individual, that is, psychological well-being. For example, “I am not particularly optimistic about the future,” “I am well satisfied about everything in my life,” “I am very happy,” “Life is good,” and “I always have a cheerful effect on others” etc. The person expresses the degree of agreement with the statements on a 6-point Likert scale (1 = *strongly disagree*; 6 = *strongly agree*). The studies carried out with a sample of people aged between 13 and 68 years verified the good reliability of this scale (α = 0.91). The Spanish adaptation with adolescents showed good internal consistency (α = 0.86), as in this study (α = 0.97).

*Test de Empatía Cognitiva y Afectiva"* [*Cognitive and Affective Empathy Test*] (TECA) [41]. This measures empathy in 4 dimensions: (1) Perspective taking: The intellectual or imaginative ability to put oneself in another’s place; (2) Emotional comprehension: the ability to recognize and understand other people’s moods, intentions, and impressions; (3) Empathetic stress: the ability to share another person’s negative emotions; and (4) Empathetic Joy: the ability to share someone else’s positive emotions. The TECA consists of 33 items on which participants rate their degree of agreement on a Likert response format ranging from 1 (*strongly disagree*) to 5 (*strongly agree*). The overall score is obtained by the sum of all the items. The test shows good reliability (α = 0.86) for the entire questionnaire, and also suitable for the sample of this study (α = 0.73). The aforementioned four subscales obtained alpha coefficients of 0.74, 0.77, 0.66, and 0.77, respectively.

### 2.3. Procedure

This study uses a descriptive and comparative cross-sectional methodology. Firstly, a letter was sent to the headmasters of the randomly selected schools, explaining the research project. Those who agreed to participate received informed consent for parents and participants. When the director of the selected center refused to collaborate, the procedure was repeated with the next center on the list, taking into account the type (public-private) of the center that declined to participate. Approximately 50% of the selected centers refused to participate. This rejection can be explained by the taboo concerning the subject of this study (bullying/cyberbullying) that still exists in many educational centers. Subsequently, the evaluation team visited the schools and administered the assessment tools to the students (in two 50-minute session). Data collection was carried out during the 2017–2018 academic year.

The study met the ethical values required in research with human beings, respecting the fundamental principles included in the Helsinki Declaration, in its latest version, and in the active rules: informed consent and right to information, protection of personal data, and guarantees of confidentiality, non-discrimination, gratuity, and the possibility of dropping out of the study in any of its phases. This study received the favorable report of the Ethics Committee of the University of the Basque Country (CEISH-UPV/EHU: M10_2017_094MR1).

### 2.4. Data Analysis

The following analyses were conducted: (1) to identify the prevalence of cyberaggressors, the frequencies and percentages of students who reported having performed one or more of the 15 evaluated cyberbullying behaviors were calculated, and contingency analyses were performed as a function of sex, obtaining Pearson’s chi square; (2) in order to analyze the role of the cyberaggressor, the sample was first categorized into 2 profiles (cyberaggressors–non-cyberaggressors) and then into 3 profiles (non-cyberaggressors = had never performed any of the 15 cyberbullying behaviors; occasional cyberaggressors = had sometimes engaged in cyberbullying behavior; severe cyberaggressors = had frequently engaged in cyberbullying behaviors). Subsequently, with each variable under study, descriptive analyses (means, standard deviations) and univariate analyses were performed, first by comparing 2 profiles (non-cyberaggressors–cyberaggressors), and then 3 profiles (non-cyberaggressors–occasional cyberaggressors–severe cyberaggressors); the effect size (η^2^) was calculated (small = 0.01; medium = 0.06; large = 0.14), and post hoc tests (Bonferroni) were carried out to compare the 3 profiles; (3) several tests were performed to explore whether cyberaggressors, compared to non-cyberaggressors, have requested more help or psychological assistance for a variety of internalizing and externalizing problems: contingency analysis between being or not being a cyberaggressor, having or not having sought psychological assistance, obtaining Pearson’s chi square; descriptive analysis and analysis of variance by comparing the score in the “cyberaggression level” indicator of those who requested psychological assistance with the level of those who did not request it; and Spearman’s correlation (Rho) between the score on cyberaggression and having or not having sought psychological assistance for various problems; and (4) multiple linear regression analyses, with the sample as a whole and also by sex, to identify the variables that predict a high score in cyberaggression. In these analyses, cyberaggression was introduced as a dependent variable and as predictor variables: bullying and cyberbullying roles (victim of bullying, bullying aggressor, cybervictim), all the psychopathological symptoms evaluated, the personality variables, and/or the adaptive variables (self-image, happiness, and empathy). Statistical analyses were carried out using the Statistical Package for the Social Sciences (SPSS) 24 (IBM®). 

## 3. Results

### 3.1. Cyberaggression: Prevalence, Sex Differences and the Most Frequent Behaviors

The results revealed 32.7% (*n* = 509) of cyberaggressors (students who claimed to have performed one or more cyberbullying behaviors over the course of their lives). Of this percentage, 27.4% (*n* = 427) were occasional cyberaggressors (they claimed to have engaged in some cyberbullying behavior), whereas 5.3% (*n* = 82) were severe cyberaggressors (they admitted having frequently engaged in cyberbullying behaviors). Furthermore, 67.3% (*n* = 1049) of the sample had never performed cyberbullying behaviors (non-cyberaggressors). The percentage of male and female cyberaggressors in the sample was: 40.6% males (*n* = 315) and 24.8% females (*n* = 194), with a significantly higher percentages of male cyberaggressors (χ^2^ = 44.11, *p* < 0.001). The five most prevalent behaviors reported by cyberaggressors were: (1) sending offensive or insulting messages via mobile or mail (22.3%); (2) making anonymous calls to frighten others (7%); (3) stealing someone’s password (6.5%); (4) attacking and/or placing the victim in a humiliating situation, recording it, and uploading the video to the internet or mobile (5.6%); and (5) phoning to say offensive or insulting things (5.4%). 

### 3.2. Cyberaggression: Connection with Other Bullying/Cyberbullying Roles with Psychopathological Symptoms, and with Personality and/or Adaptive Variables

In order to confirm possible differences between the 2 profiles (non-cyberaggressors–cyberaggressors) in the different variables under study, a multivariate analysis of variance (MANOVA) based on the profile with the scores of all the variables as a whole was performed. The results indicated that there were significant differences depending on the profile, Wilks’ Lambda Λ = 0.786, *F*(21, 1442) = 18.70, *p* < 0.001, with a large effect size, η^2^ = 0.214. The results of the descriptive analyses and analyses of variance between the two profiles (see Table 1) showed that, compared to non-cyberaggressors, cyberaggressors had significantly: (1) higher scores on the indicators of victimization and perpetration of face-to-face bullying, and also on cybervictimization; (2) more psychopathological symptoms (depression, somatization, obsession-compulsion, interpersonal sensitivity, anxiety, hostility, paranoid ideation, and psychoticism), and increased overall level of psychopathology (although no differences were found in social anxiety and phobic anxiety); (3) higher scores in self-image associated with masculine characteristics and lower scores in self-image associated with feminine attributes; (4) lower level of happiness; and (5) lower overall empathy and in three of its four dimensions (perspective-taking capacity, empathetic stress, and empathetic joy; no differences were found in emotional comprehension).

In addition, 3 cyberaggression profiles (non-cyberaggressors, occasional cyberaggressors, severe cyberaggressors) were compared by performing a MANOVA as a function of the profile with the scores in all the variables. The results indicated significant differences as a function of the profile, Wilks’ Lambda Λ = 0.793, *F*(42, 2882) = 11.21, *p* < 0.001, with a large effect size, η^2^ = 0.140. The results of the descriptive analyses and analyses of variance (see Table 2) revealed significant differences between the 3 profiles, respectively, in 4 variables: Compared to non-cyberaggressors, occasional cyberaggressors performed significantly more face-to-face aggressions, suffered more cybervictimization, had less empathetic joy, and less global empathy; and, in turn, compared to casual cyberaggressors, severe cyberaggressors had carried out significantly more face-to-face aggressions, had suffered more cybervictimization, had less empathetic joy, and less overall empathy. In the rest of the variables, the relevant differentiation occurred mainly between being or not being a cyberaggressor, as there were few nuances between being an occasional or a severe cyberaggressor. Thus, as shown when analyzing the sample in 2 profiles, we found that, compared to non-cyberaggressors, occasional cyberaggressors had also been more frequently victims of face-to-face bullying, they showed more psychopathological symptoms (depression, somatization, obsession-compulsion, interpersonal sensitivity, anxiety, hostility, paranoid ideation, and psychoticism), and a higher overall level of psychopathology, a high self-image of masculinity and a low one of femininity, lower capacity of perspective taking, and empathetic stress. In the variable happiness, we found that non-cyberaggressors were happier than severe cyberaggressors, but no differences were found between non-cyberaggressors and occasional cyberaggressors. In addition, the absence of a connection between cyberaggression and symptoms of social anxiety, phobic anxiety, and emotional understanding was confirmed. 

### 3.3. Cyberaggressors and Demand for Psychological Assistance

In the variable “demand for psychological assistance”, in the sample as a whole, 73.9% (*n* = 1151) had never consulted a psychologist, whereas 24.7% (*n* = 385) claimed to have consulted a psychologist for various reasons (anxiety, sadness, academic performance…). Another 1.4% did not respond.

The contingency analysis carried out between being or not being a cyberaggressor and having or not having receiving psychological assistance showed that 33% of the 509 cyberaggressors (*n* = 168; one third) had requested psychological assistance whereas only 20.7% of the 1049 non-cyberaggressors (*n* = 217) had requested assistance. Pearson’s chi-square yielded significant differences (χ^2^ = 28.36, *p* < 0.001), showing that a higher proportion of cyberaggressors than of non-cyberaggressors had requested psychological assistance. 

To determine differences in the level of cyberaggression between those who had requested psychological assistance and those who had not requested it, an analysis of variance (ANOVA) was calculated with the scores on cyberaggression as a function of this condition (psychological assistance). The results showed that the mean score in cyberaggression in those who had gone to a psychologist was significantly higher (*M* = 1.18, *SD* = 2.30), than that of those who had never gone to a psychologist (*M* = 0.75, *SD* = 2.50), *F*(2, 1556) = 4.48, *p* < 0.05 (very small effect size, η^2^ = 0.006, *r* = 0.07). In short, participants who had consulted a psychologist for various symptoms had significantly higher scores in cyberaggression (amount of perpetrated cyberbullying behavior) compared to those who had never gone to a psychologist. Also the correlation coefficient (Spearman’s ρ) showed that the higher the score in cyberaggression, the greater the likelihood of having sought psychological assistance (*r_s_* = 0.14, *p* < 0.001).

### 3.4. Cyberaggression: Predictor Variables

To examine Hypothesis 4, multiple regression analysis was carried out, using cyberaggression as the criterion variable and all the previously described variables as predictors. Stepwise linear regression analyses identified the variables that predicted cyberaggression. These four predictors were introduced in stepwise hierarchical regression, and the results are presented in Table 3 (general sample), Table 4 (boys), and Table 5 (girls). No additional or control variables were included. 

As can be seen, out of the set of variables, four were predictors of cyberaggression for the total sample: cybervictimization (β = 0.26), bullying aggression (β = 0.17), empathetic joy (β = −0.12), and self-image of masculinity (β = 0.05). Therefore, four variables, which explained 14.9% of the variance, were predictors of cyberaggression: having been a cybervictim, having perpetrated many bullying behaviors, having low empathetic joy, and a self-image based on attributes associated with masculinity. When analyzing the sample by sex, we confirmed the same variables in the girls, which explained 18.4% of the variance, whereas in boys, only two variables showed significant coefficients: having been a cybervictim and low empathetic joy; bullying aggression showed a marginal coefficient and masculinity self-image did not add explained variance. These variables explained 16.7% of the variance.

## 4. Discussion

The study had four objectives: (1) to calculate the prevalence of cyberaggressors in a sample of adolescents of Bolivia; (2) to compare non-cyberaggressors with cyberaggressors in other bullying/cyberbullying roles, in psychopathological symptoms, and in self-image of masculinity/femininity, happiness, and empathy; (3) to analyze whether cyberaggressors consulted with a psychologist more than non-cyberaggressors; and (4) to identify predictor variables of cyberaggression. 

Firstly, the results confirm a prevalence of 32.7% of cyberaggressors (27.4% occasional and 5.3% severe), with a higher percentage of males. These results confirm Hypothesis 1, although it slightly exceeds the predicted percentage, and point in the same direction as those found in a systematic review of studies that reported a range of cyberaggressors between 2.5% and 32% [2]. The only study conducted in Bolivia that provides information on cyberaggressors [3] found 12%. The discrepancies can be explained by the different ages of the samples in that study (many were 12–13 years old) and the sample from this study (14–15 years), and, as has been shown in various studies [29,42], from ages 12–13 to 14–15 there is an increase in cyberbullying. In addition, this study, like many others [1,4,5,6], finds a higher percentage of male cyberaggressors. However, another study [43], showed a high variation in prevalence based on the type of cyberaggression analysed. Verbal cyberaggression and online exclusion were more common than impersonation and visual cyberaggression. They also found there were generally no statistically significant differences between boys and girls. Nevertheless, when differences do appear, boys generally tend to be more aggressive than girls, while girls are more likely to be victims. 

Secondly, compared to non-cyberaggressors, cyberaggressors had engaged in significantly more face-to-face bullying behaviors, had suffered more face-to-face victimization and cybervictimization behaviors, they had more psychopathological symptoms (depression, somatization, obsession-compulsion, interpersonal sensitivity, anxiety, hostility, paranoid ideation, and psychoticism) and a higher overall level of psychopathology, they self-defined through many characteristics associated with masculinity and few associated with femininity, they had lower feelings of happiness, as well as less overall empathy in three of its four dimensions (less cognitive ability of perspective taking, less ability to share others’ negative and positive emotions). These results confirm Hypothesis 2 almost entirely, but no differences were found between non-cyberaggressors and cyberaggressors in symptoms of social anxiety and phobic anxiety, or in emotional understanding. 

These results point in the direction of other studies that have found connections of cyberaggression with face-to-face aggression [23] and relational aggression [21]. In addition, an association was identified between being a cyberaggressor and being a cybervictim [28,44]. The results also confirm the significant overlap between bullying roles and the different forms of bullying (victim, aggressor, cybervictim, cyberaggressor). In addition, they emphasize the importance of carrying out multidirectional anti-bullying/cyberbullying activities with groups of adolescents, emphasizing all the behaviors that are suffered and performed in bullying (cyberaggression, face-to-face aggression, face-to-face victimization and cybervictimization). 

On the other hand, the findings of this study also confirm other studies that have found connections between cyberaggression and psychopathology. The results of this study confirm the research that has found that cyberaggressors have more symptoms of depression and anxiety [8,9,15], stress [10,14,15], feelings of loneliness [11,12,15], psychosomatic problems, and psychopathological symptoms in general [13]. Cyberaggressors have many psychopathological symptoms of various kinds, and, although causal relationships cannot be assumed because this is a cross-sectional study, this suggests that cyberaggression is related to a high level of psychological distress.

There are no studies that have explored the self-image of cyberaggressors associated with gender stereotypes (masculinity–femininity), so the findings of this study represent a contribution to our knowledge. Cyberaggressors have been defined significantly with many adjectives associated with masculinity (athletic, strong personality, desire to take risks/loves danger, dominant, aggressive, acts as leader, individualistic, hard, and selfish), characteristics that emphasize achievement, materialism, and competition (instrumental people), and with very few adjectives associated with femininity (affectionate, sensitive to others’ needs, understanding, compassionate, warm, tender, child-loving, weeps easily, submissive), characteristics emphasize affective sharing, quality of life, and interpersonal harmony (expressive people). These findings suggest a relevant connection between different types of violence, in particular, between peer violence and sexist violence closely associated with stereotypes of masculinity. It is important to note that instrumentality or masculinity predicts cyberaggression only in the case of females, who usually score lower than males and show a large variability in this dimension, and a trend towards increased scores in this dimension [45]. However, these results should be interpreted with caution because, although BSRI is an instrument that has been widely used, it has some limitations, as there are different ways of conceiving masculinity and femininity (linear versus orthogonal). Although the BSRI is a self-descriptive scale, the categorization of masculinity and femininity and gender stereotypes is limited. Recently, the study of Ferrer-Pérez and Bosch-Fiol [46] showed that, regardless of gender and sex typing, most people considered that only some of the items of the BSRI describe men or women characteristically, while the rest could apply equally to both.

In relation to happiness, some studies have associated being a cyberaggressor with lower optimism and happiness [16] and less satisfaction with life [15], which is convergent with the results obtained in this study. The lower level of happiness of cyberaggressors may be related to the high level of psychological distress due to various psychopathological symptoms that they suffer, as has been seen.

Regarding empathy, the results confirm studies in which cyberaggressors, compared to non-cyberaggressors, had less empathy [17,18], but research analyzing cognitive empathy and affective empathy separately show varied and sometimes contradictory results. Renati et al. [19] only found lower emotional empathy; in the study of Rodríguez-Hidalgo and collaborators, low affective and cognitive empathy were predictors of cyberaggression [20], whereas in our study, we found less ability of perspective taking (cognitive empathy), and less ability to share another person’s negative and positive emotions (affective empathy), but we failed to find less emotional comprehension (ability to recognize and understand others’ moods) (cognitive empathy). The discrepancies may be explained by the different instruments used to measure empathy and the different ages of the samples of the studies.

Thirdly, we confirmed that cyberaggressors had sought more psychological assistance than non-cyberaggressors, although just over one third of the cyberaggressors had gone to a psychologist. However, compared to those who had never consulted a psychologist, participants who had requested psychological assistance for various symptoms had significantly higher scores on the cyberaggression indicator (amount of perpetrated cyberbullying behavior). Also converging, the correlations confirm that the higher the score in cyberaggression, the more likely the individual is to have sought psychological assistance. These results, in addition to confirming Hypothesis 3, also highlight that cyberaggressors have more psychological problems for which they request psychological assistance, which connects cyberaggression, psychological distress, and psychopathology, through a measure that complements the SCL-90-R. The results confirm those obtained by Garaigordobil and Machimbarrena [14] with children aged 10 to 12 in which it was also found that cyberaggressors had consulted a psychologist significantly more frequently than those not involved in cyberbullying situations.

Finally, four predictor variables of cyberaggression are identified: having been a cybervictim, having been a face-to-face aggressor (bullying), having low empathetic joy, and a self-image based on attributes associated with masculinity. The results confirm Hypothesis 4 almost entirely, as only one dimension of empathy, low empathetic joy, was shown to be a predictor. In addition, this study finds that cyberaggressors’ self-image associated with masculinity is also a predictor, which had not been hypothesized, given the absence of studies with this variable. The results confirm other studies that have found that antisocial behavior (traditional bullying and rule breaking) [22], and having been a perpetrator of face-to-face bullying predict cyberaggression [24,25,26]. The meta-analysis of Guo [27] also confirmed this, emphasizing that having been a bully was a robust predictor. In addition, having been cybervictimized was also a predictor of cyberaggression [20,24].

This study has some implications for clinical contexts because the results show the connections between cyberaggression and psychopathology, as well as the overlapping of roles in bullying/cyberbullying (cyberaggressors, cybervictims, aggressors, and victims). All this emphasizes the importance of therapeutic intervention with all those involved in bullying/cyberbullying situations. Clinical intervention should focus on reducing psychopathological symptoms and promoting psychological well-being, empathy, and happiness. 

In addition, the results of the study have practical implications for the prevention of bullying/cyberbullying in educational contexts. The school can play a very important role in prevention, performing two functions: (1) Identifying/evaluating those involved; and (2) Implementing antibullying programs to encourage empathy for victims and rejection of this type of violence. In schools, all those involved in bullying/cyberbullying situations should be identified. Self-reports that can identify victims, cybervictims, aggressors, and cyberaggressors could include, for example: The School Violence Questionnaire–Revised [47]; Cyberbullying. Screening of peer harassment. Screening of face-to-face school bullying and technological bullying (cyberbullying) [29,30]; The Cybervictimization Questionnaire [48]; The Cyberaggression Questionnaire [49]; or the Triangulated Cyberbullying Questionnaire [50]. 

In addition, the school must implement prevention/intervention programs of antibullying, which contain activities to confront students with this type of violence (bullying/cyberbullying), promoting empathy and positive socio-emotional behaviors (prosocial behavior, sensitivity to the needs of others, compassion, affective warmth etc.). Among the psychoeducational intervention programs to prevent bullying/cyberbullying, we include: The CIP program (Concienciar Informar Prevenir) to raise awareness, report and prevent bullying [51]; the ConRed program to build and coexist on the Internet and in the social media [52]; the Program of the Child’s Ombudsman in the Community of Madrid [53]; the Cyberprogram 2.0 to prevent cyberbullying [54,55,56]; Bullying: Psychoeducational Intervention Guide [57]; the Anti-Bullying Project [58]; the Prev@cib Program of intervention for prevention of bullying and cyberbullying [59], or the proposals presented in Campbell and Bauman’s book [60]; However, few antibullying programs include specific components for vulnerable populations (LGBT –Lesbian, Gay, Bisexual and Transgender–, obese, disabled, gifted, minority racial groups...), although recent studies have shown the increased suffering and impact of victimization/cybervictimization of the people in these groups [61,62].

However, as has been shown in previous works [63], to prevent bullying in any of its modalities, psychoeducational interventions in school are not sufficient, as the family context is very relevant for the development and maintenance of positive social behavior, as well as for the prevention and confrontation of bullying/cyberbullying. Therefore, it is important to encourage parental education about childrearing guidelines that reduce the likelihood of a child becoming a victim or an aggressor. For example, stimulating children’s self-esteem, recognizing and reinforcing spontaneous positive behaviors, demonstrating unconditional love, promoting empathy and prosocial behavior, setting limits without authoritarianism, dedicating time to the children, being a good model of behavior, making communication a priority at home. [1]

This work provides relevant data, making a contribution to our knowledge, but it is not without limitations: (1) the use of self-reports due to the social bias involved; and (2) as it is a cross-sectional study, causal relationships cannot be established between the variables.

## 5. Conclusions

The percentage of adolescent cyberaggressors in Cochabamba (Bolivia) is worthy of consideration, as 32.7% reported having performed one or more cyberbullying behaviors towards other classmates, and of them, 5.3% admitted having performed them very frequently. A higher percentage of male cyberaggressors was found. The most common behaviors identified were sending offensive or insulting messages via mobile or email, and password theft.

Compared to non-cyberaggressors, cyberaggressors engaged in more face-to-face bullying, had suffered more victimization and cybervictimization, had more psychopathological symptoms, a higher overall level of psychopathology, had requested more psychological assistance for various symptoms, and had a self-image based more on attributes of masculinity, lower feelings of happiness, and less capacity for empathy.

Being or having been a cybervictim, being or having been a face-to-face bully, having low empathetic joy, and a self-image based on attributes of masculinity were predictors of cyberaggression.

The findings emphasize the need for therapeutic intervention with all those involved in bullying/cyberbullying situations, in order to reduce their psychopathological symptoms. In addition, the results suggest that, from educational contexts, actions should be taken to identify/evaluate bullying/cyberbullying and to prevent/intervene in this type of violence through anti-bullying programs.

Among the lines of research indicated by the findings of this study, we suggest the implementation in school contexts of programs to prevent and reduce bullying/cyberbullying behaviors, evaluating their effects through the reduction of psychopathological symptoms and through the increase of feelings of happiness.

## Figures and Tables

**Table 1 ijerph-17-01022-t001:** Means and standard deviations in all variables in the two profiles (non-cyberaggressor, cyberaggressor), results of analyses of variance as a function of profile, and effect size (η^2^).

Variables	Non-Cyberaggressors (*n* = 1049)	Cyberaggressors (*n* = 509)	*F* (1, 1555)	*p*	η^2^
	*M (SD)*	*M (SD)*			
Bullying/Cyberbullying					
Bullying victimization	1.55 (1.99)	2.20 (1.95)	35.79	0.000	0.024
Bullying aggression	0.93 (1.44)	2.36 (2.18)	214.45	0.000	0.128
Cybervictimization	1.49 (2.75)	3.95 (4.30)	180.98	0.000	0.110
Psychopathological symptoms					
BDI-II Depression	10.75 (10.04)	13.96 (10.65)	33.99	0.000	0.023
SAS Social anxiety	46.00 (17.56)	46.86 (15.75)	1.41	0.234	0.001
SCL90 Somatization	0.85 (0.85)	1.07 (0.88)	20.30	0.000	0.014
SCL90 Obsession-compulsion	1.13 (0.98)	1.34 (0.97)	15.94	0.000	0.011
SCL90 Interpersonal sensitivity	0.90 (0.89)	1.07 (0.93)	12.88	0.000	0.009
SCL90 Depression	0.97 (0.89)	1.10 (0.89)	8.00	0.005	0.005
SCL90 Anxiety	0.89 (0.98)	1.03 (0.93)	8.88	0.003	0.006
SCL90 Hostility	0.80 (0.93)	1.10 (0.98)	34.07	0.000	0.023
SCL90 Phobic anxiety	0.78 (0.91)	0.83 (0.90)	0.84	0.358	0.001
SCL90 Paranoid ideation	0.78 (0.88)	0.96 (0.88)	14.28	0.000	0.010
SCL90 Psychoticism	0.77 (0.86)	0.96 (0.89)	16.49	0.000	0.011
SCL90 GSI	0.89 (0.75)	1.06 (0.77)	18.31	0.000	0.012
Personality variables					
BSRI Masculinity self-image	34.22 (10.76)	36.75 (10.44)	17.47	0.000	0.012
BSRI Femininity self-image	38.40 (12.02)	36.21 (10.88)	11.11	0.001	0.008
OHQ Happiness	114.54 (35.53)	110.85 (34.41)	4.03	0.045	0.003
TECA Perspective-taking	27.15 (4.55)	26.12 (4.53)	14.66	0.000	0.010
TECA Emotional comprehension	28.98 (4.68)	29.22 (4.71)	0.93	0.333	0.001
TECA Empathetic stress	25.92 (4.75)	24.99 (4.62)	11.83	0.001	0.008
TECA Empathetic joy	30.26 (5.62)	28.93 (5.57)	18.46	0.000	0.012
TECA Overall empathy	112.31 (14.12)	109.26 (13.26)	14.76	0.000	0.010

Notes: GSI = General Symptomatic Index; *M* = Mean, *SD* = Standard deviation, *F* = Fisher’s F, *p* = Significance, η^2^ = Eta-squared effect size.

**Table 2 ijerph-17-01022-t002:** Means and standard deviations in all the variables of the three profiles (non-cyberaggressor, occasional, severe cyberaggressor), results of the analyses of variance as a function of the profile, effect size (η^2^), and post hoc group comparison (Bonferroni).

Variables	Non-Cyberaggressor (*n* = 1049)	Occasional Cyberaggressor (*n* = 427)	Severe Cyberaggressor (*n* = 82)	*F* (1, 1555)	*p*	η^2^	Post-hoc
	*M (DT)*	*M (DT)*	*M (DT)*				
Bullying/Cyberbullying							
Bullying victimization	1.55 (1.99)	2.27 (2.06)	2.17 (1.83)	20.72	0.000	0.026	1 < 2.3
Bullying aggression	0.93 (1.44)	2.23 (2.02)	3.40 (3.09)	140.85	0.000	0.153	1 < 2 < 3
Cybervictimization	1.49 (2.75)	3.62 (3.84)	6.22 (6.04)	121.79	0.000	0.135	1 < 2 < 3
Psychopathological symptoms							
BDI-II Depression	10.75 (10.04)	13.57 (16.14)	16.14 (12.19)	18.586	0.000	0.024	1 < 2.3
SAS Social anxiety	46.00 (17.56)	46.89 (15.62)	47.84 (17.58)	0.75	0.473	0.001	-
SCL90 Somatization	0.85 (0.85)	1.07 (0.86)	1.08 (0.95)	10.55	0.000	0.014	1 < 2
SCL90 Obsession-com	1.13 (0.98)	1.35 (0.97)	1.31 (0.98)	8.39	0.000	0.011	1 < 2
SCL90 Interper. sensiti.	0.90 (0.89)	1.09 (0.94)	1.05 (0.94)	6.84	0.001	0.009	1 < 2
SCL90 Depression	0.97 (0.89)	1.10 (0.86)	1.18 (1.08)	4.44	0.012	0.006	1 < 2
SCL90 Anxiety	0.89 (0.98)	1.04 (0.92)	1.10 (1.08)	4.84	0.008	0.006	1 < 2
SCL90 Hostility	0.80 (0.93)	1.09 (0.96)	1.19 (1.14)	18.09	0.000	0.023	1 < 2.3
SCL90 Phobic anxiety	0.78 (0.91)	0.83 (0.91)	0.92 (1.01)	1.03	0.355	0.001	-
SCL90 Paranoid ideat.	0.78 (0.88)	0.98 (0.89)	0.98 (0.98)	8.32	0.000	0.011	1 < 2
SCL90 Psychoticism	0.77 (0.86)	0.97 (0.89)	1.05 (1.04)	9.25	0.000	0.012	1 < 2.3
SCL90 GSI	0.89 (0.75)	1.07 (0.77)	1.11 (0.88)	9.91	0.000	0.013	1 < 2.3
Personality variables							
BSRI Masculinity	34.22 (10.76)	36.41 (10.12)	39.35 (11.81)	13.12	0.000	0.017	1 < 2
BSRI Femininity	38.40 (12.02)	36.48 (11.02)	34.94 (11.12)	6.36	0.002	0.008	1 > 2
OHQ Happiness	114.54(35.53)	110.91 (34.41)	104.39 (37.73)	4.16	0.016	0.005	1 > 3
TECA Perspective-tak.	27.15 (4.55)	26.19 (4.50)	25.34 (4.61)	11.14	0.000	0.014	1 > 2.3
TECA Emotional comp.	28.98 (4.68)	29.30 (4.72)	28.74 (4.71)	0.895	0.409	0.001	-
TECA Empathet. stress	25.92 (4.75)	25.08 (4.74)	24.41 (4.26)	7.55	0.001	0.010	1 > 2.3
TECA Empathetic joy	30.26 (5.62)	29.37 (5.39)	26.02 (5.65)	23.71	0.000	0.030	1 > 2 > 3
TECA Overall empathy	112.31(14.12)	109.94 (13.29)	104.52 (14.49)	14.71	0.000	0.019	1 > 2 > 3

Notes: Obsession-com = Obsession-compulsion; Interper. sensiti. = Interpersonal sensitivity; Paranoid ideat. = Paranoid ideation; GSI = General Symptomatic Index; *M* = Mean, *SD* = Standard deviation, *F* = Fisher’s *F*, *p* = Significance, η^2^ = Eta-squared effect size. Post hoc = Group comparison (Bonferroni): 1 = Non-cyberaggressor; 2 = Occasional cyberaggressor; 3 = Severe cyberaggressor.

**Table 3 ijerph-17-01022-t003:** Predictor variables of cyberaggression for the whole sample.

Variables	Model 1			Model 2			Model 3			Model 4		
	*B*	*SE*	β	*B*	*SE*	β	*B*	*SE*	β	*B*	*SE*	β
Cybervictimization	0.215	0.016	0.316 ***	0.182	0.017	0.266 ***	0.179	0.017	0.263 ***	0.177	0.017	0.260 ***
Bullying aggression				0.253	0.032	0.196 ***	0.230	0.032	0.178 ***	0.222	0.032	0.172 ***
Empathetic joy							−0.050	0.010	−0.115 ***	−0.051	0.010	−0.117 ***
Masculinity self-image										0.011	0.005	0.047 *
Adjusted *R*^2^	0.099			0.135			0.147			0.149		
Δ R^2^	-			0.036			0.013			0.002		
*F* for change in *R*^2^	170.645 ***			64.207 ***			23.324 ***			3.982 *		

Note: *B* = Beta; *SE* = Standard error; β = standardized beta; Δ *R*^2^ = Change in *R*^2^; * *p* < 0.05. *** *p* < 0.001.

**Table 4 ijerph-17-01022-t004:** Predictor variables of cyberaggression for boys.

Variables	Model 1			Model 2			Model 3			Model 4		
	*B*	*SE*	β	*B*	*SE*	β	*B*	*SE*	β	*B*	*SE*	β
Cybervictimization	0.333	0.029	0.387 ***	0.313	0.030	0.364 ***	0.308	0.030	0.358 ***	0.307	0.030	0.356 ***
Bullying aggression				0.125	0.053	0.082 *	0.103	0.052	0.068	0.100	0.053	0.066
Empathetic joy							−0.068	0.018	−0.0123 ***	−0.069	0.018	−0.125 ***
Masculinity self-image										0.007	0.009	0.025
Adjusted *R*^2^	0.149			0.154			0.168			0.167		
Δ R^2^	-			0.006			0.015			0.001		
*F* for change in *R*^2^	135.081 ***			5.663 *			13.628 ***			0.583		

Note: *B* = Beta; *SE* = Standard error; β = standardized beta; Δ R^2^ = Change in *R*^2^; * *p* < 0.05. *** *p* < 0.001.

**Table 5 ijerph-17-01022-t005:** Predictor variables of cyberaggression for girls.

Variables	Model 1			Model 2			Model 3			Model 4		
	*B*	*SE*	β	*B*	*SE*	β	*B*	*SE*	β	*B*	*SE*	β
Cybervictimization	0.127	0.017	0.266 ***	0.087	0.016	0.182 ***	0.085	0.016	0.177 ***	0.082	0.16	0.171 ***
Bullying aggression				0.334	0.033	0.337 ***	0.324	0.034	0.327 ***	0.314	0.034	0.316 ***
Empathetic joy							−0.024	.010	−0.077 *	−0.025	0.010	−0.081 *
Masculinity self-image										0.012	0.006	0.069 *
Adjusted *R*^2^	0.070			0.175			0.180			0.184		
Δ R^2^	-			0.107			0.006			0.005		
*F* for change in *R*^2^	58.919 ***			99.968 ***			5.426 *			4.392 *		

Note: *B* = Beta; *SE* = Standard error; β = standardized beta; Δ R^2^ = Change in R^2^; * *p* < 0.05. *** *p* < 0.001.

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
