# Peer review of "Cyberaggression in Adolescents of Bolivia: Connection with Psychopathological Symptoms, Adaptive and Predictor Variables"

_ijerph, 2020, doi:10.3390/ijerph17031022_

Round 1

Reviewer 1 Report

This is a useful article with data from an under-studied population. It will benefit from some fairly minor revision.

Abstract: mention near the start that the study is in Bolivia.

4 aims are given, but only 3 findings – the (3) on line 27 refers to aim (4).

Introduction

Lines 62-63 a possible useful reference here is:

Smith, P.K., López-Castro, L., Robinson, S. & Görzig, A. (2019). Consistency of gender differences in bullying in different cross-cultural surveys. Aggression and Violent Behavior, 45, 33-40.

Line 113. Do we need a Hypothesis here?  H1 especially seems superfluous – why not just aim to find out the prevalence in this Bolivian sample (and perhaps compare with other samples?).

Really I do not see the need for H2, H3 or H4 to be stated as hypotheses, either – although they are reasonable expectations, it would be perfectly okay to state these as ‘finding out’ objectives.

Materials and methods

Somewhere – perhaps in Procedure around line 236 – the date (year) when the survey was carried out, should be stated. While important generally, this is especially so when studying cyberbullying!

Line 225 – explain more the procedure around ‘randomly selected schools’ – are they from just Cochabamba – is that a city in Bolivia? What population? How many schools in total from which the18 took part? Did many schools refuse to take part?

Results

Tables 1 and 2 repeat a lot of data – specifically, the left hand column labels and the next column of Non-cyberaggressors data, is identical – there is the possibility of combing them. Under the F value column, 2 dp is quite enough, e.g. 35.795 could be 35.80.

Lines 339-349 – the findings on requesting psychological assistance might be best in a table (maybe similar format to Table 2).

Discussion

Lines 372-375 – the findings are 32.7% cyberaggressors, and this is compared with a range from previous studies of 2.5% to 32% - so is H1 confirmed? Or disconfirmed? (it is outside the previous, very wide, range!). This show the dubious nature of calling this a Hypothesis; plus, the prevalence rates depend not only on sample (e.g. age, as mentioned) but also definition given, time reference period, frequency criterion used, etc – almost any prevalence rate can be obtained, depending on the kind of question asked …

Around p.477 – interventions for cyberbullying – the author(s) may consider citing the recent book on this:

Campbell & S. Bauman (eds.), Reducing cyberbullying in schools. London: Elsevier.

Author Response

Reviewer 1.

Abstract: Line 18 states that they are students from Bolivia. The study presents 4 objectives, because we wanted to differentiate psychopathological symptoms evaluated with a self-report of a behavioral indicator such as consulting a psychologist for problems (contrasting the results with two different methods – self-reported symptoms versus psychological aid to treat problems). However, in view of the results, in order to decrease redundancy and increase clarity, we have arranged the results in three blocks, integrating in the second block the findings that cyberaggressors present more psychopathological symptoms in the self-report, which is also evident in their higher demand for psychological assistance.

Introduction: The reference suggested by the reviewer has been included (line 63) (in references: nr. 60).  We do not understand the reason for eliminating all the hypotheses, as their formulation is the norm in all studies. APA norms indicate their inclusion as a relevant field of any investigation. It is true that some researchers question formulating hypotheses in prevalence studies, but in this case, following the systematic review of the prevalence literature in Latin America and specifically in Bolivia, a hypothesis is formulated, providing an expected percentage, based on the percentages obtained in previous studies with a similar Latin American population. In our study, we formulated a hypothesis, we obtained the prevalence within the sample, and in the discussion, we compared it with the prevalences of other studies in Bolivia and Latin America. We appreciate the reflection but we consider it useful to maintain the hypotheses based on the literature related to each one. However, following the reviewer's suggestion, information is added to provide further foundation to Hypothesis 1 (line: 113) and the wording of the hypotheses has been improved (lines from 113 to 126).

Procedure: The requested information about the date of data collection has been included (lines 234-235). The description of the sample (in Participants) indicates that they are adolescents from Cochabamba (Bolivia) and that the sample is representative, providing the total N of students of this educational level (31,895) from which the representative sample was obtained. In relation to the issues raised by the reviewer, the requested information (Cochabamba population, total of the educational centers...) has been added. This information requested by the reviewer is now included in the Participants section (line 131). As reported in the MS, 18 public and private centers were chosen at random, and when one of the selected centers rejected collaboration, another center with similar characteristics (public-private) was selected. The schools' level of rejection to participate in this study was high (approximately 50%, as we contacted 40 centers to select the 18 that collaborated). Following the reviewer's suggestion, now in the procedure section, new information about the rejection percentage has been included (lines from 231 to 233).

Results: In Table 1, as suggested, the third decimal place has been removed from the F values. The reviewer is right, the average scores and standard deviations of non-cyberaggressors are repeated in Tables 1 and 2. However, including all the information in a single table would be complex due to the number of columns that are needed. The final width of the table would be excessive and would be a problem to edit. For this reason, the differentiation between non-cyberaggressor and cyberaggressor is maintained in Table 1, and in Table 2, the difference between occasional and severe is maintained.  Regarding how to present information about psychological assistance, it can be done in two formats, either by means of a table or by describing it in the text. In this case you, we chose to describe the results in three lines of text in order to reduce the length of the MS, as the article has 5 tables.

Discussion: In fact, the hypothesized percentage is confirmed, although it slightly exceeds the predicted percentage. Now this is has been qualified in the discussion (lines 389-390). In fact, epidemiological studies are difficult to compare, given the different variables involved in each research (different instruments, specific behaviors evaluated, time considered in the evaluation....), and indeed, the studies sometimes do not provide information on differentiating behaviors that are performed once or more times (global) or frequently (severe), making it more difficult to compare the studies. As suggested, the quotation of Campbell and Bauman's book (line 512) is now included (in references: nr. 61).

Reviewer 2 Report

General Comments for authors: Overall, t the authors detail out an interesting study regarding about cyberaggression in adolescents. In the discussion, implications for clinical contexts and practical implications for the prevention of bullying/cyberbullying in educational contexts are very interesting. Theoretical and diversified references are correctly used. However, just some adjustments are requested: Introduction. The authors provide a detailed account of the theoretical background of the cyberaggression literature. However, the authors said: “Cyberbullying poses a threat to mental health and psychological well-being during childhood, adolescence, and youth. Cybervictims suffer feelings of insecurity, loneliness, sadness, unhappiness, helplessness, anxiety, irritability, depression, suicidal ideation (some commit suicide), post-traumatic stress, fear, low self-esteem, anger and frustration, somatizations, sleep disorders, eating disorders, phobias, academic performance issues... In addition, these consequences are …”. Please expand upon the references and/or clarify the existing references. Measuring Instruments. a) Why some instruments include information about the reliability of these Spanish versions and not others? If you decide to include this information, please include information about the reliability of these Spanish versions of these measures and information about whether these measures have been validated in adolescent populations. b) The authors are asked to work to balance how reliability is presented. Choose, adjust and standardize this. c) Some instruments describe the temporality of the instrument and others do not. Later, the authrors said: “The results revealed 32.7% (n = 509) of cyberaggressors (students who claimed to have performed one or more cyberbullying behaviors over the course of their lives)”. This was probably the reason for the high percentage. Please clarify this in your discussion. Hypotheses. In the hypotheses 1, the authors said: “ It is expected to find approximately 30% of cyberaggressors (who have performed one or more cyberbullying behaviors) of whom 5% will be severe (have very frequently performed cyberbullying behaviors). Why 30 %? Before, the authors said: “The systematic review of studies carried out on the prevalence of bullying and cyber-bullying in Latin America published between 2005 and 2018 [2] showed a high preva-lence of cyberaggressors, whose percentages in the different studies range from 2.5% to 32%”. It is for this reason? If so, justify it. Discussion. The authors said: “There are no studies that have explored the self-image of cyberaggressors associated with Gender stereotypes (masculinity-femininity), so the findings of this study represent a contribution to our knowledge. Cyberaggressors have been defined significantly with many adjectives associated with masculinity (athletic, strong personality, desire to take risks/loves danger, dominant, aggressive, acts as leader, individualistic, hard, and selfish), characteristics that emphasize achievement, materialism, and competition (instrumental people), and with very few adjectives associated with femininity (affectionate, sensitive to others' needs, understanding, compassionate, warm, tender, child-loving, weeps easily, submissive), characteristics emphasize affective sharing, quality of life, and interpersonal harmony (expressive people). These findings suggest a relevant connection between different types of violence, in particular, between peer violence and sexist violence closely associated with stereotypes of masculinity”. The scale used in this paper provides a contribution to our knowledge, but it is not without limitations. Sex Role Inventory (BSRI) has some limitations that should be considered. it has been and continues to be a widely-used instrument of psychological testing. According to this framework, the masculinity–femininity construct is considered a masculine dimension, consisting of a masculine–non-masculine continuum, and a feminine one, consisting of a feminine–non-feminine continuum. Sexually-typed persons are those who have interiorized the socially-established sexual standards, attributing to themselves those traits considered typical of their sex to a high degree and excluding those considered typical of the other; androgynous persons are those to which traits considered masculine and feminine are simultaneously attributed to a high degree (Ferrer- Pérez & Bosch-Fiol, 2014). However, it would be very interesting and relevant to include in the limitations of the study that there are different ways of conceiving masculinity and femininity. Although the BSRI, it is a self-descriptive scale. The categorization of masculinity and femininity, and gender stereotypes is limited. From this perspective, self-categorization, in other words, whether a person defines himself or herself as masculinity and femininity, androginity,…..is particularly relevant. Why has the category of undefined or androgenic not been included in the analysis? Is there a very low percentage? If so, it should be included in the limitations. General. • Choose, adjust and standardize between “Cyberaggression” or “cyberbullying behavior”. It is important to always use the same expression everywhere (article’s title, table’s title, in the tables, in the text, etc.). • Possible typo in line 233: “guarant ees “ References: Ferrer-Pérez, V.A, & Bosch-Fiol, E. (2014). The Measure of the Masculinity-Femininity Construct Today: Some Reflections on the Case of the Bem Sex Role Inventory. Journal of Social Psychology, 29 (1), 180–207.

Author Response

Reviewer 2

Introduction: The statements are based on the conclusions presented in the review carried out by Garaigordobil in 2018. Many authors could be quoted whose results confirm the serious consequences of cyberbullying for victims and aggressors. Among others Cross, Lester y Barnes, 2015; Chang et al., 2013; Garaigordobil, 2013; González-Cabrera et al., 2017; Gradinger, Strohmeier, y Spiel, 2009; Heiman et al., 2015; Hinduja y Patchin, 2010; Iranzo, Buelga, Cava, y Ortega-Barón, 2019; Kowalski, Limber y Agatston, 2010; Landoll et al., 2015; McLoughlin, Burgess y Meyricke, 2009; Mitchell, Ybarra y Finkelhor, 2007; Navarro et al., 2015; Schneider et al., 2012; Sourander et al.,  2010; Tsitsika et al., 2015; Vieno et al., 2014; Wright, 2016; Ybarra, Diener-West, y Leaf, 2007. All these studies show the negative effects at the emotional, psychosocial, academic level... All these works are referenced in the review of Garaigordobil of 2018 but, in order not to over-expand the references in the article (currently 63), the review is quoted for the readers who wish to deepen their knowledge.

Instruments: All instruments had included the reliability of the test in the original sample of the test and also the reliability obtained with the sample of our study, which is the usual procedure. All instruments are validated in Spanish and for adolescent population or for student population with broader age ranges. In original Spanish instruments (Screening of Peer Harassment & TECA), reliability is that of the original sample of the instrument with adolescents and young people. In line with the reviewer's request, in the tests that are adaptations, we have now included the coefficients of Spanish adaptations with adolescents of similar ages to our sample (SAS-A: lines 172-173; OHQ: lines 213-214), and with students with broader age ranges (BDI-II: lines 166-167; BSRI: lines 203-204), as many Spanish adaptations contain samples with broader age ranges, and no specific information of the coefficient with adolescents is provided. The temporality is indicated in those instruments that include this information in the application instructions. Where no information is provided, it is because the test does not report the temporality in the instructions. The information provided in the article is indicated for each instrument.

Results (hypothesis 1): 32.7% reported having performed cyberbullying one or more times. Considering the severity that a single behavior of cyberbullying may have (for example, a scene of "happy slapping") the percentage is very relevant. It may even be more relevant than the same global percentage of face-to-face bullying, because in cyberspace, everything is recorded and it is almost impossible to erase the traces of the victims' aggression/humiliation. However, in the study, we also differentiate occasional and severe, finding 5.3% of very frequent cyberaggressors, which is very serious. Therefore, we do consider it important to identify even those who have bullied just once (27.4%) because a single behavior can have a serious emotional impact on the cybervictim. In addition, 30% was also found in some Latin American studies identified in the systematic review conducted by Garaigordobil et al. (2018). In these studies, we find a range of cyberaggressors between 2.5% and 32%, and this range can be partly explained by the fact that, in some studies, the percentage is of cyberaggressors who have performed one or more behaviors, whereas others provide information about severe cyberaggressors, although researchers do not always report this clearly. For this reason we maintain the formulation of the hypothesis on expected prevalence, we rely on Latin American studies, and subsequently contrast the hypothesis by comparing the prevalence found with other studies in Latin America. Following the reviewer's suggestion, information is added to provide further foundation to Hypothesis 1 (line: 113)

Discussion: As the reviewer suggests, a paragraph referring to the limitations of the BSRI and its concept of masculinity-femininity has been included in the discussion. In addition, the reference suggested by the reviewer has been incorporated, adding more discussion in relation to this point (lines: from 439 to 448) (inn references: nr. 63). In relation to the use of terms, cyberaggression-cyberaggressors: Cyberaggression is used to refer to the amount of cyberaggression behavior (indicator with continuous direct score) and cyberaggressor to designate the person who performs it (percentage of students who perform cyberaggression behaviors). The text has been reviewed, and we hope that it is now clear. About the typo, in the MS, we indicate guarantees.

Reviewer 3 Report

Manuscript title: „ Cyberaggression in adolescents of Bolivia: Connection with psychopathological symptoms, adaptive and predictor variables “
Manuscript number: 714388
Thank you for the opportunity to review this manuscript. It describes the prevalence, co-occurrence with face-to-face bullying, and risk factors of cyberaggression.
1. Theory – From a psychologist perspective, the manuscript lacks of theory and theoretical contribution.
2. Objectives and hypotheses – It is unusual to state hypotheses about prevalence rates.
Please formulate hypothesis in present tense, e.g.: “Cyberaggressors consulte psychologists significantly more frequently than those who do not engage in cyberbullying behaviors towards other classmates.”
3. Data analysis – I would not write profiles as you make no profile analysis but only categorize the students into two classes.
Please also provide a justification why you used analyses that assume continuous variables and linear relationships (i.e., analysis of variance, regression analysis) but then calculated the non-parametric Spearman’s rho instead of the parametric Pearson’s r correlation?
4. Results – Please also report the percentages of the most prevalent cyberaggression behaviors.
The results section should be written in past tense
The regression results in Table 3 are not well described. Which of the variables presented in Table 3 were included first? Did you include any control variables that were not reported in the Table? The change in R2 is also not comprehensible. In the regression analyses separated by sex, represent the regression coefficients the full model (with all variables included) or the model with the two variables (for the boys) and four variables (for the girls), respectively.
5. Discussion – you should also include a short outlook paragraph. What would be interesting for future research to study, given the findings in the present study.

Author Response

Reviewer 3

Theory: Information on the theoretical framework of the study has now been included before the study's 4 objectives (lines: 100-101) are mentioned. This work makes a contribution with practical implications for educational and clinical contexts, rather than theoretical contributions.

Hypothesis: It is true that some researchers question formulating hypotheses in prevalence studies, but in this case, following the systematic review of the prevalence literature in Latin America and specifically in Bolivia, a hypothesis is formulated, providing an expected percentage, based on the percentages obtained in previous studies with a similar Latin American population. In our study, we formulated a hypothesis, we obtained the prevalence within the sample, and in the discussion, we compared it with the prevalences of other studies in Bolivia and Latin America. We appreciate the reflection, but we consider it useful to maintain the prevalence hypothesis based on the reviewed literature related to it. With regard to redacting the hypotheses in present tense, we note that Hypothesis 3 has been reworded according to the suggestion of the reviewer. Hypotheses 2 and 4 have been formulated in present tense, and Hypothesis 1 has been contextualized without making further changes (lines from 113 to 126).

Data Analysis: The profiles and their explanation in this section are important to be able to better understand the subsequent analyses based on these previous categorizations. We analyze the characteristics of each group or profile (non-cyberaggressors, cyberaggressors -occasional and severe) in a diversity of variables, some continuous quantitative (symptoms, happiness, empathy….) and other categories (being or not being a cyberaggressor, having or not having consulted a psychologist). The analyses performed are correct and vary depending on the type of variable we used (continuous or categorical in their different combinations). For example: (1) to know whether there are differences between non-cyberaggressors and cyberaggressors (occasional and severe) in various variables (symptoms, empathy, happiness….), we use categorical variables (being or not being a cyberaggressor) and continuous direct scores in variables (symptoms, empathy, happiness….). Therefore, after performing previous analyses, means and standard deviations are obtained in each variable, as well as the results of the variance and its significance when comparing these groups; (2) When we relate 2 categorical variables (having or not having consulted a psychologist and having or not having performed one or more cyberbullying behaviors), we perform contingency analyses, obtaining Pearson's chi square; (3) However, when we work with a categorical variable (whether or not psychological assistance was requested) and a continuous direct score in the cyberaggression indicator (amount of cyberbullying behavior performed), we use Spearman's correlation... We hope that this explanation clarifies the question raised.

Results: The revised version of the MS includes the information requested in relation to the percentages of the most prevalent cyberaggression behaviors (lines 281-284). Now, throughout the results section, the past tense is used, changing some verbs that were in present tense to past tense (lines from 270 to 371). Regression: Results in Tables 3, 4 & 5 are now well described (see new two paragraphs). Variables were chosen based on a previous stepwise analysis in the general sample. No additional control variables were included. Full model with all variables are included in the new tables. Change in R2 is described as well as F change (lines: from 357 - 371). In place of the former Table 3, Tables 3, 4 and 5 have been prepared and present the information requested by the reviewer. We hope everything is now correct.

Discussion: Now, at the end of the article, we incorporated a paragraph about the lines of research opened by the findings of this study (lines: from 548 to 551)

Reviewer 4 Report

The relevance of the study of cyber aggression is due to the significant severity of this problem in cyberspace and its consequences for the subjective well-being of the individual and interpersonal relations. This study attempts to analyze psychological characteristics of adolescents depending on the manifestations of cyber aggression (presence and absence) of adolescents. The authors identified a fairly large number (32.7%) of cyber-aggressive adolescents in the sample, and in different series of studies revealed similarities and differences between cyberaggressors and non-cyberaggressors.  The value of the study lies in the fact that the authors have established the psychological problems of aggressive adolescents, including their behavioral (bullying and victimization), personal and emotional characteristics.  The results of the study were thoroughly discussed, conclusions were made, and recommendations on therapeutic intervention were proposed. The research presented in the article is scientifically significant, has an important social significance and is recommended for publication.

Wish to the authors. It is necessary to justify the choice of a method for comparing different parts of the sample. In table 3, the authors present R2 indicators instead of ΔR2. It would be better to present ΔR2= R2i - R2j that is, the difference between the subsequent and previous indicator R2, below to provide a General indicator R2 and indicator of the Fisher test (Statistical significance of R according to the Fisher criterion) with a significance level, such as (R2=0.15; F=32.5 p<0.01).

Author Response

Revisor 4

Regresión: Los resultados en las Tablas 3, 4 y 5 ahora están bien descritos (ver dos párrafos nuevos). Las variables fueron elegidas en base a un análisis previo por etapas en la muestra general. No se incluyeron variables de control adicionales. El modelo completo con todas las variables se incluye en las nuevas tablas. Se describe el cambio en R 2 , así como el cambio F (líneas: de 357 a 371). En lugar de la anterior Tabla 3, las Tablas 3, 4 y 5 se han preparado y presentan la información solicitada por el revisor. Esperamos que todo esté correcto ahora.